# CFTR Modulators Rescue the Activity of CFTR in Colonoids Expressing the Complex Allele p.[R74W;V201M;D1270N]/dele22_24

**DOI:** 10.3390/ijms24065199

**Published:** 2023-03-08

**Authors:** Karina Kleinfelder, Elena Somenza, Alessia Farinazzo, Jessica Conti, Virginia Lotti, Roberta Valeria Latorre, Luca Rodella, Arianna Massella, Francesco Tomba, Marina Bertini, Claudio Sorio, Paola Melotti

**Affiliations:** 1Department of Medicine, Division of General Pathology, University of Verona, Strada Le Grazie 8, 37134 Verona, Italy; 2Endoscopic Surgery Unit, Azienda Ospedaliera Universitaria Integrata Verona, 37126 Verona, Italy; 3Cystic Fibrosis Centre, Azienda Ospedaliera Universitaria Integrata Verona, Piazzale Stefani, 1, 37126 Verona, Italy

**Keywords:** rectal organoids, complex CFTR alleles, personalized medicine, CFTR modulators, Ussing chambers, FIS assay

## Abstract

An Italian, 46-year-old female patient carrying the complex allele p.[R74W;V201M;D1270N] in trans with CFTR dele22_24 was diagnosed at the Cystic Fibrosis (CF) Center of Verona as being affected by CF-pancreatic sufficient (CF-PS) in 2021. The variant V201M has unknown significance, while both of the other variants of this complex allele have variable clinical consequences, according to the CFTR2 database, with reported clinical benefits for treatment with ivacaftor + tezacaftor and ivacaftor + tezacaftor + elexacaftor in patients carrying the R74W-D1270N complex allele, which are currently approved (in USA, not yet in Italy). She was previously followed up by pneumologists in northern Italy because of frequent bronchitis, hemoptysis, recurrent rhinitis, *Pseudomonas aeruginosa* lung colonization, bronchiectasis/atelectasis, bronchial arterial embolization and moderately compromised lung function (FEV1: 62%). Following a sweat test with borderline results, she was referred to the Verona CF Center where she presented abnormal values in both optical beta-adrenergic sweat tests and intestinal current measurement (ICM). These results were consistent with a diagnosis of CF. CFTR function analyses were also performed in vitro by forskolin-induced swelling (FIS) assay and short-circuit currents (Isc) in the monolayers of the rectal organoids. Both of these assays showed significantly increased CFTR activity following treatment with the CFTR modulators. Western-blot analysis revealed increased fully glycosylated CFTR protein after treatment with correctors, in line with the functional analysis. Interestingly, tezacaftor, together with elexacaftor, rescued the total organoid area under steady-state conditions, even in the absence of the CFTR agonist forskolin. In conclusion, in ex vivo and in vitro assays, we measured a residual function that was significantly enhanced by in vitro incubation with CFTR modulators, especially by ivacaftor + tezacaftor + elexacaftor, suggesting this combination as a potentially optimal treatment for this case.

## 1. Introduction

Cystic fibrosis (CF) is a severe, progressive disease that is caused by loss-of-function mutations of the cystic fibrosis transmembrane conductance regulator (CFTR) gene. It is the most common autosomal, recessive monogenic disorder amongst the European population [1]. A defective CFTR channel impairs the normal epithelial fluid secretion in several organs, compromising their functions. Among the affected organs, lung disease is the major cause of morbidity and mortality in CF subjects. The loss of the critical hydration of the airway surface liquid and mucus layer leads to recurrent respiratory infections and to early obstructive lung disease and, eventually, to respiratory failure [2]. So far, over 2100 mutations have been identified in CF patients, however, less than 30% of them were annotated in the CFTR2 database as disease-causing mutations (www.cftr2.org (accessed on 20 December 2022)). Each pathogenic variant can affect distinct, single, or multiple steps of the CFTR biogenesis and activity. The association of several variations in cis (at the same allele, known as a complex allele) can change the CFTR function. Moreover, the combination of different pathogenic variants (in trans) may worsen the clinical situation of CF patients, due to additive defects. Thus, complex alleles can represent a further source of CFTR genetic variability, clinical phenotype and variable response to the CFTR modulators [3,4,5,6].

The effect of additional mutations in cis, combined with the effect of intragenic variability on CFTR expression and function, is difficult to predict and reproduce in cell line model systems, with the final impact on CFTR function being more properly investigated in patient-derived samples [4]. In this scenario, data from functional studies [7,8] help us to assess the pathogenicity of these variants and to predict the in vivo response to modulator therapies at an individual level [9,10]. Here, we report a not-yet characterized complex allele [R74W;V201M;D1270N] that is associated with CF in an Italian patient. The complex allele [R74W;D1270N] (HGVS nomenclature: c.[220C > T; 3808G > A]) has been reported as disease-causing, being found in a Moroccan CF patient [11]. In another case of an infertile man with CBAVD carrying the P841R mutation, it was present on the other allele [12]. On the other hand, the variant V201M seems to be non-benign, being associated with CFTR-related disorders, according to CFTR-France, which has registered 45 patients with this mutation (https://cftr.iurc.montp.inserm.fr/cftr (accessed on 20 January 2023)), whereas the CFTR2 database (cftr2.org) has only recorded 14 subjects harboring this variant, with an uncertain clinical impact. No functional characterization nor theratyping studies have been reported for this complex allele. The CFTR dele22_24 variant is disease-causing when it is combined with another CF-causing variant. It removes exons 22 to 24, corresponding to exons 25 to 27 in the new numbering, causing the loss of part of the NBD2 domain and the stop codon, including the poly (A) signal [13]. This important deletion produces a major defect in the CFTR protein that cannot be recovered by the currently available CFTR correction strategies. In this study, we use patient-derived rectal organoids, which are a well-established in vitro model, following the N-of-1 trial principle, to evaluate the potential response of the complex allele to FDA-approved therapeutic options.

## 2. Case Presentation

### 2.1. Clinical Features of a CF Subject Carrying p.[R74W;V201M;D1270N]/dele22_24-CFTR Mutations and the Functional Response In Vivo and Ex Vivo

This is a case of a female patient carrying the complex allele p.[R74W;V201M;D1270N], which was found in trans with a type 1 mutation (CFTR dele22_24), with a current age of 46, who was followed up by pneumologists because of frequent bronchitis, bronchial arterial embolization followed by hemoptysis, recurrent rhinitis, *Pseudomonas aeruginosa* lung colonization and bronchiectasis/atelectasis. The patient was referred to the CF Center of Verona in 2021, where she had borderline values of sweat tests, according to the Gibson and Cooke method (with chloride 43 mmol/L, Na:Cl ratio 0.96) and a compromised pulmonary function (FEV1 62% of predicted value).

In order to better characterize the combination of these variants, we investigated the functional response of p.[R74W;V201M;D1270N]/CFTR dele22_24 in rectal biopsies by intestinal current measurements (ICM) in Ussing chambers. ICM are a quantitative biomarker of the CFTR activity in the intestinal epithelium and are used to reveal CFTR dysfunction [14]. Rectal biopsies from non-CF subjects yield a strong positive deflection in the short-circuit current (Isc) in response to secretagogues, reflecting mainly electrogenic chloride secretion. The intestinal biopsy from the patient with p.[R74W;V201M;D1270N]/CFTR dele22_24 mutations yielded an upward deflection after the addition of compounds (carbachol, forskolin/IBMX and histamine) with a lower magnitude of responses, suggesting residual CFTR function. The patient’s ICM values, ΔIsccarb + cAMP + Hista 59 ± 30 µA/cm^2^, fit below the normal range that was registered for a non-CF group (Figure 1A,B).

To assess the CFTR function in vivo, we performed the optical β-adrenergic sweat test, which is a sensitive assay that is able to discriminate CF from carriers and non-CF subjects [15,16]. It is known that, in CF patients, the cholinergic pathway for the secretion of eccrine sweat glands works normally, whilst β-adrenergic sweating is compromised [17]. The rate limiting for β-adrenergic sweat secretion reflects the half-normal β-adrenergic sweat rates that are seen in CF carriers when the rates are expressed as a ratio of cholinergic sweating [18]. Hence, we could identify 115 ± 23 individual sweat glands in a selected region on the forearm of the CF patient (p.[R74W;V201M;D1270N]/CFTR dele22_24) after 10 min of methacholine injection (M-sweat; CFTR-independent sweating). From these, only 4 ± 1, representing 3 ± 1% of the identified sweat glands, responded to the cocktail injection of isoproterenol + aminophylline + atropine (C-sweat; CFTR-dependent sweat), with an average mean C/M ratio of 0.000273 ± 0.0003; with such value falling in the CF group (Figure 2A–C). The nasal potential difference was not measured in this case because of the recurrent rhinitis.

### 2.2. Theratyping of p.[R74W;V201M;D1270N]/dele22_24-CFTR in Rectal Organoids

We then evaluated whether this CFTR genotype could benefit from the currently available CFTR-targeted therapies. We, therefore, developed intestinal organoids from the rectal biopsies that were collected for the ICM procedure to predict the in vivo benefit by first performing a FIS assay [19]. Three correctors and one potentiator, which are commercially available and have been approved for clinical use, were selected to investigate their capacity to induce functional restoration in this case. The R74W-V201M-D1270N/dele22_24-CFTR organoids presented a residual function, as first observed in our ICM assessment, that can be further increased with the use of type I and III corrector agents. Treatment with VX-770 (ivacaftor) alone, or together with VX-809 (lumacaftor), VX-661 (tezacaftor) and VX-661-VX-445 (elexacaftor), increased the FIS of these organoids up to three-fold of AUC values that were seen for the baseline at 0.128µM fsk#x2014;a concentration that corresponds well with the in vivo assessment of the CFTR function and clinical effects [20,21]. It is worth noting that the FIS rates in those organoids that were pre-incubated with VX-661 and VX-661-VX-445 were underestimated, due to their partially recovered steady-state lumen area prior to the FIS assay. In the absence of fsk, VX-809 pretreatment is unable to restore the lumen area of the rectal organoids, as previously described [20]. The pharmacological recovery of the lumen area is indicative of drug efficacy. Hence, for a proper evaluation of the drug effect, it is appropriate to add the value that is registered in the drug rescued-organoids (steady-state total organoid area (SOA) condition) to the swelling values that are obtained by the FIS assay, thus allowing us to estimate the maximal restoration of the CFTR activity in R74W-V201M-D1270N/dele22_24 organoids for the CFTR modulators that have been tested (Figure 3A,B).

The WT-CFTR function cannot be properly estimated by the FIS assay due to a swollen phenotype [8,22]. In order to extend the results that were obtained by FIS assay and also to compare the mutant CFTR with healthy CFTR function, we measured the agonist-induced CFTR activity in organoid-derived monolayers (colonoids) by short-circuit current (Isc) analysis. The treatments with CFTR modulators led to a significant increase in FSK-stimulated currents, calculated to be up to 4-fold higher than of the basal Isc that was registered for untreated colonoids. The results of the analyses were as follows: vehicle-ΔIsc: 3.1 ± 3.3 µA/cm^2^, VX-770-ΔIsc: 4.3 ± 3.7 µA/cm^2^; VX-809 + 770-ΔIsc: 8.8 ± 5.4 µA/cm^2^, VX-661 + 770-ΔIsc: 12.8 ± 10.4 µA/cm^2^; and VX-445 + 661 + 770-ΔIsc: 13.1 ± 10.7 µA/cm^2^. Thus, in presence of modulators, the mutant CFTR channel could reach a maximum of approximately 11% of the WT activity that was recorded in our setting (Figure 4A,B).

Afterward, we evaluated the impact of these mutations on CFTR protein folding or maturation and processing rescued by the correctors. The effect of VX-809, VX-661 and VX-661 together with VX-445 on the expression of R74W-V201M-D1270N-CFTR, the core-glycosylated protein (band B) and the mature fully glycosylated protein (band C) were evaluated using Western blotting. As a control, organoids were treated with the vehicle, dimethyl sulfoxide (DMSO). In the absence of modulators, the mature protein, visible as band C, was barely detectable. The VX-809 treatment did not induce an appreciable increase, while VX-661 alone provided a slight increase in band C, though it was still below a significance threshold at the densitometric analysis. However, the double-corrector treatment (VX-661-VX-445) induced a strong band C expression, in line with the increased functional response that was recorded (Figure 5A,B). Given these experimental results, the positive response to the modulators that we detected was likely due to the increased protein expression associated with the complex R74W-V201M-D1270N-CFTR allele. Indeed, CFTRdele22_24 consists of an important deletion of 9454 bp, including exons 25 to 27 (22 to 24), the stop codon and the poly (A) signal [13], producing a major defect in the CFTR protein that cannot be recovered by the CFTR modulators that have been tested.

### 2.3. Materials and Methods

#### 2.3.1. Ethics Statement

Written informed consent was obtained from all subjects according to the local ethical committee’s rules (CRCFC-CFTR028).

#### 2.3.2. CFTR Modulators

The CFTR modulators VX-770, VX-809 and VX-661 were purchased from Selleckchem, whereas VX-445 was purchased from MedChemExpress (Monmouth Junction, NJ, USA)

#### 2.3.3. Intestinal Current Measurement (ICM)

Superficial rectal mucosa samples (2–4 per donor) were freshly obtained using biopsy forceps and placed in cold DMEM/F12 media (Gibco) with antibiotics (50 µg/mL gentamicin and 50 µg/mL vancomycin). The intestinal current measurements were performed under voltage-clamp conditions, according to the standard operating procedure of the European Cystic Fibrosis Society ICM-SOP (version 2.7, 2011: https://www.ecfs.eu (accessed on 12 February 2012)). The transepithelial Isc values (short-circuit currents, Isc) were recorded in a system of recirculating Ussing chambers (P2250) (Physiologic Instruments), using multi-channel voltage clamp EVC4000 (World Precision Instruments, Sarasota, FL, USA). Briefly, the biopsy tissues were mounted in sliders (P2407C) of Ussing chambers and bathed in a symmetrical Meyler saline solution (pH 7.4) containing (mM) the following: 10 mM Hepes, 0.3 Na_2_HPO_4_, 0.4 NaH_2_PO_4_, 1.0 MgCl_2_, 1.3 CaCl_2_, 4.7 KCl, 128 NaCl, 20.2 NaHCO_3_, 10 mM D-glucose and 0.01 indomethacin. The Meyler solutions were continuously gassed with 95% O_2_–5% CO_2_ and maintained at 37 °C. The basal potential difference (PD basal), short-circuit current (Isc, basal) and transepithelial resistance (Rt, basal) were determined and the tissues were stabilized for 40 min. The Isc, as a direct measure for the net movement of ions across the epithelium, was registered after adding specific compounds to the mucosal (M) or serosal (S) bathing solutions as specified: forskolin (FSK, 0.01 mM, M + S) + IBMX (0.1 mM, M + S), to activate cAMP-dependent Cl^−^ (e.g., CFTR channel) and ORCC (outwardly rectifying Cl^−^ channels) channels; carbachol (0.1 mM, S), to start the Cl^−^ secretion linked to the cholinergic Ca_2_^+^ and protein kinase C; DIDS (0.2 mM, M), to inhibit DIDS-sensitive, non-CFTR Cl^−^ channels; histamine (0.5 mM, S), to reactivate Ca_2_^+^-dependent secretion and to measure the DIDS-insensitive components of Ca_2_^+^-dependent Cl^−^ secretion. The ICM tracings were recorded with PowerLab (8/35, AD Instruments, Bella Vista, Australia) and were analyzed using Lab Chart v8 software (MLS060/8, AD Instruments, Bella Vista, Australia). The reference values of the ICM diagnostic parameters were developed at the CF Center of Verona. We decided to use the cumulative chloride secretory response ΔIsc, Forsk/IBMX, ΔIsc, carbachol and ΔIsc and histamine, which has been shown to represent the most conclusive diagnostic ICM parameter to differentiate patients with questionable CF into CF and non-CF [23]. Here, we used the outcome of ICM in a subgroup of 24 non-CF healthy controls (ΔIsccarb + cAMP + Hista 197 ± 66 µA/cm^2^) and a subgroup of 29 CF patients as the cut-off value ΔIsccarb + cAMP + Hista 80 µA/cm^2^ to discriminate between these two groups.

#### 2.3.4. Intestinal Organoids Culture from Rectal Biopsies

Human intestinal organoids derived from patients were obtained from a rectal biopsy after the isolation of crypts containing adult stem cells. In short, four to six rectal biopsies were stored at 4 °C in a surgical medium, washed with cold PBS solution and incubated with 10 mM EDTA for 90–120 min at 4 °C. The supernatant was harvested and the EDTA was washed away. The crypts were isolated by centrifugation and embedded in Matrigel (growth factor reduced, phenol red-free; Corning) and seeded (50–100 crypts per 40 μL Matrigel per well) in 24-well plates. After an incubation period of 15–30 min at 37 °C, the plated crypts were immersed in a pre-warmed complete culture medium consisting of advanced DMEM/F12 supplemented with penicillin and streptomycin, 10 mM hepes, Glutamax, N2, B27 (all from Invitrogen), 1 μM N-acetylcysteine (Sigma) and the following growth factors: 50 ng mL^−1^ mouse epidermal growth factor (mEGF), 50% Wnt3a-conditioned medium (WCM) and 10% noggin-conditioned medium (NCM), 20% Rspo1-conditioned medium, 10 mM nicotinamide (Sigma), 10 nM gastrin (Sigma), 500 nM A83-01 (Tocris) and 10 μM SB202190 (Sigma). The complete medium supplemented with 10 µM Rho inhibitor (Y27623) and 10 µM Chir (Chir90221) (both from Sigma) and additional antibiotics, gentamycin and vancomycin (both added 1:1000), was used only for the first culture week. The medium was refreshed every 2–3 days and the outgrowing crypts/organoids were expanded 1:3–1:5 times every 7–10 days.

#### 2.3.5. Evaluation of Drug Recovery of Stead-State Total Organoid Area (SOA) Assay

The rectal organoids from a 7- to 10-day-old culture were disrupted mechanically and seeded in a 96-well plate in 5 μL of 50% Matrigel (Corning) containing 20–30 organoids and immersed in 50 μL of culture medium with one of the following treatments: DMSO (0.1%), 3 µM VX-809, 3 µM VX-661, or 3 µM VX-445, or their combination. After 2 h of seeding and after 20–24 h of treatment, the treated organoids were analyzed with fluorescence microscopy by taking one picture at t = 0 hpt (0 h post treatment) and another picture at t = 24 hpt (24 h post treatment) (EVOS Cell Imaging System, Thermo Fisher Scientific, Waltham, MA, USA) with 4× objective, at 37 °C and 5% CO_2_. The total area (xy plane) of the majority of organoids in a well was analyzed and the SOA of the selected organoids was calculated manually using ImageJ and GraphPad Prism version 7 (GraphPad Software, San Diego, CA, USA). The SOA was expressed as the area under the curve (AUC t = 24 h; baseline, 100%). Occasionally, cell debris and nonviable structures were excluded from the analysis.

#### 2.3.6. Forskolin-Induced Swelling (FIS) Assay

CFTR function recovery was then evaluated by performing a FIS assay. For the analysis, the organoids were stimulated with forskolin concentrations of 0.128 μM and were directly analyzed with fluorescent microscopy (EVOS Cell Imaging System, Thermo Fisher Scientific) with 4x objective, at 37 °C and 5% CO_2_, every 30 min for a total acquisition of 120 min in a time-lapse video. Every condition was analyzed in duplicate (two wells were used to study one condition). For the CFTR correction, the organoids were pre-incubated for 24 h with 3 µM VX-661, 3µM VX-809 and 3µM VX-445, or combinations thereof. For CFTR potentiation, 3 μM VX-770 was added simultaneously with forskolin. The area of the same set of organoids that were evaluated for SOA (plane xy) was evaluated related to t = 0 and to FSK treatment manually, as follows: the organoids were numbered progressively, using PowerPoint software, creating a mask. The subsequent overlapping of the mask to the images ensured the evaluation of the same set of organoids over time. Then, the circumference of each numbered organoid was measured using ImageJ software [24] and a freehand selection tool and the corresponding area was automatically calculated in pixels by the software. The normalized data are expressed as the total area under the curve (AUC, t = 120 min; baseline, 100%), which was calculated using GraphPad Prism version 7 (GraphPad Software, San Diego, CA, USA).

#### 2.3.7. Organoid-Derived Monolayers Cultures

For the culture of the epithelial monolayers, seven-day-old, extracellular matrix-embedded, intestinal organoids were suspended in advanced DMEM (4 °C; Gibco) and washed by centrifugation (5 min, 1500× *g*) to remove the matrix. The intestinal organoids were dissociated by a brief (45 s, 37 °C) incubation in trypsin (0.25%) solution (Gibco), followed by mechanical disruption through a 200 µL pipette tip (Greiner). Such a step was repeated for the number of times necessary to obtain single-cell suspension. Then, 250,000 cells were seeded for each insert (filter 6.5 mm insert; Ref. 3470, Costar) with a 100 μL organoid culture medium added on top and 600 μL of organoid culture was added on the bottom side of the pre-coated filters with collagen type IV from human placenta (10 µg/cm^2^) (234,154, Sigma) diluted in saline phosphate buffer. The colonoids were cultured in a 5% CO_2_ atmosphere at 37 °C. The culture medium was supplemented with Y27632-Rho and CHIR-99021 (both from Sigma-Aldrich) during the first two days after seeding. The organoid culture medium (without supplement) was changed every other day after a minimum of 7–10 days of cell culture.

#### 2.3.8. Transepithelial Electrical Resistance (TEER)

The formation of the organoid monolayers was monitored by morphologic observation using an Olympus CKK31 inverted microscope (Olympus, Japan). The TEER was measured using an EVOM2 epithelial volt ohmmeter (World Precision Instruments) before refreshing the medium. The readings of the volt ohmmeter can be multiplied by the surface area of the Transwell inserts (0.33 cm^2^) to calculate the unit area of resistance (Ω.cm^2^). A TEER value of 300–400 Ω.cm^2^ was considered an index of complete monolayer formation.

#### 2.3.9. Short-Circuit Measurements

The electrophysiological measurements were performed directly on the filter using specific Ussing chambers (P2300) and sliders (P2302T) (Physiologic Instruments). The transepithelial voltage was clamped at 0mV (EVC4000 multi-channel voltage/current clamp, World Precision Instruments, Sarasota, FL, USA) after compensating for voltage offsets and current. For the measurement of the anion secretion mediated by CFTR on the colonoids, the chambers containing a 2D filter were filled with a symmetrical Meyler saline solution (pH 7.4) (10 mM Hepes; 0.3 mM Na_2_HPO_4_; 0.4 mM NaH_2_PO4; 1.0 mM MgCl_2_; 1.3 mM CaCl_2_; 4.7 mM KCl; 128 mM NaCl; 20.2 mM NaHCO_3_; and 10 mM D-glucose), kept at 37 °C and constantly gassed with carboxygen (95% O_2_, 5% CO_2_). The filters were then tested with the following components, which act positively in CFTR activity: 10 µM forskolin (Sigma) in both apical and basolateral sides (AP + BL) and 0.3 µM VX-770 (AP + BL). The experiment was concluded with the addition of the CFTR inhibitor, 20 µM PPQ-102 (Tocris), from the apical and basolateral sides. The tracks were recorded with PowerLab (8/35, AD Instruments, Bella Vista, Australia) and were analyzed with Lab Chart v8 software (MLS060/8, AD Instruments, Bella Vista, Australia) in accordance with the standard operating protocol.

#### 2.3.10. Immunoblotting

The untreated 3D organoids, or those treated with CFTR correctors, were disrupted with advanced DMEM/F12 supplemented with 1% glutamax, 10 mM Hepes, 0.2% primocin and 1% penicillin/streptomycin. The Matrigel matrix was removed from the three-dimensional structures with cell recovery solution (Corning) following the manufacturer’s instructions. The pellet was lysed with 50 µL of RIPA/EDTA/DTT/vanadate lysis buffer (50 mM Tris, pH 7.5, 150 mM NaCl, 1% Triton X-100, 1% sodium deoxycholate, 0.1% SDS, 1 mM EDTA, 1 mM DTT and 1 mM sodium orthovanadate) with 1x× protease inhibitor cocktail (Roche) and then disrupted by vigorous pipetting with a P200 tip or by vortexing. After 30 min at 4 °C under agitation, the lysate was spun at 13,000× rpm at 4 °C (maximum radius of 8.5 cm; Biofuge Fresco, Heraeus, Hanau, Germany) and the supernatant was collected and subjected to protein concentration analysis (Bradford). Then, 60 µg of the total protein was further solubilized in 4x Laemmli sample buffer (4% SDS, 5% β-mercaptoethanol, 20% glycerol, 0.0025% bromophenol blue, 0.16 M Tris-HCl and pH 6.8), with 125 mM DTT and incubated at 37 °C for 20 min. The proteins were resolved by SDS–polyacrylamide gel electrophoresis using 7.5% acrylamide gels. The proteins were wet-transferred from the gel to a polyvinylidene difluoride (PVDF) membrane by electrophoresis with a transfer system (Bio-Rad) at 100 V (constant) for 60 min. The membrane was blocked with 5% non-fat dry milk/Tris-buffered saline (TBS)–Tween (0.3%) at 25 °C for 1 h, then incubated with α-CFTR monoclonal antibodies (mAb) 450, 570 and 596 (CFTR Antibody Distribution Program, Cystic Fibrosis Foundation, UNC-Chapel Hill) (1:1000 dilution each) at 4 °C overnight. The membrane was washed four times for 10 min each time with TBS–Tween (0.3%) and then incubated with horseradish peroxidase-conjugated secondary antibody α-mouse at 1:12,000 (Cell Signaling, 7076) at 25 °C for 1 h. The membrane was washed again four times for 10 min each time with TBS–Tween (0.3%) and enhanced chemiluminescence development by Westar Supernova ECL substrate detection (Cyanagen). The relative chemiluminescence intensity of the target protein was normalized to the chemiluminescence intensity of the β-actin detected by the α-beta-actin antibody (1:1000) (Cell Signaling, 4970). ImageJ software (version 1.8.0112, National Institute of Health, Bethesda, 144 MD, USA) was used to analyze the band densities and the results were confirmed using a minimum of three independent experiments.

#### 2.3.11. Optical Beta-Adrenergic Sweat Test (OBAS Test)

The image-based sweat measurements were performed using the procedure described by Wine and colleagues (2013) [15], with modified equipment. We used a Canon EOS 550D digital SLR camera equipped with a macro lens (SP AF90mmF/2.8 Di Macro 1:1, Tamron, Japan) and an annular flash (Macro EM-140 DG E0-ETTLII electronic flash). The sweat secretion rates were measured by changes in the volume of sweat drops secreted on the forearm in an oil layer, including the presence of a water-soluble blue dye (erioglaucine disodium crystals, CAS 3844-45-9, also known as Brilliant Blue FCF, FD&C Blue No.1, Acid Blue 9, E133) to improve the detection of the sweat bubbles during the two distinct phases of sweat secretion (M and C) that were induced by the stimulation of the sweat glands. The M phase lasts for 10 min and allows the CFTR-independent sweat that is produced following the intradermal injection of methacholine chloride in a specific region on the patient’s forearm to be measured. The C phase, lasting 30 min, allows the measurement of the CFTR-dependent sweat that is produced following the intradermal injection of a cocktail of β-adrenergic drugs in the same area of the forearm. Methacholine stimulates the sweat glands through the cholinergic pathway; for this reason, in the first phase of the test, a response to stimulation is observed with the subsequent production of sweat in healthy subjects, carriers and in CF subjects. The cocktail of β-adrenergic drugs instead allows the production of CFTR-dependent sweat, which is impaired in CF patients. This phase, therefore, allows for the discrimination of healthy subjects, carriers and CF patients and also allows the highlighting of the individual variability between the subjects. Atropine is included in it to inhibit the effects of the methacholine that was previously administered. Hence, we computed a ratio between the CFTR-dependent and CFTR-independent sweat secretion rates by multiple individual glands. The increase in sweat volume from each identified gland was measured over time for each patient analyzed. For the measurement of sweat secretion in both phases, pictures were analyzed manually using the program ImageJ to measure the diameter of each bubble and automatically calculate their volume. The reference values of the optical ratiometric rate sweat tests were developed at the CF Center of Verona. We decided to use the average value of the mean C/M-sweat and also the percentage of glands producing C-sweat [14,15] obtained from a subgroup of non-CF (n = 43), healthy carriers (HC; n = 44) and CF patients (n = 46).

#### 2.3.12. Gibson and Cooke Sweat Test (GCST)

The sweat was collected and the chloride concentration was determined according to the Gibson and Cooke method (GCST), following the procedure utilized routinely at the CF Center of Verona (recommendations from the SIFC working group, September 2017: www.sifc.it (accessed on 15 March 2018)).

#### 2.3.13. Statistics

The results are presented as mean ± SEM, or mean ± SD, with the number of experiments indicated. The statistical analysis was performed using parametric or non-parametric tests using GraphPad (GraphPad Software, San Diego, CA, USA) for the representative FIS and TCM assays, making comparisons between DMSO and CFTR modulators for organoid theratyping; *p* < 0.05 was considered to be statistically significant.

## 3. Discussion

In this study, we have reported the complex allele [R74W;V201M;D1270N] in trans with a class I CFTR mutation in a female patient (age 46) with a recent diagnosis of CF, having borderline values for the gold-standard sweat chloride test. The [R74W;V201M; D1270N] is a CF complex allele of uncertain clinical significance. It is known that the double-mutant allele, R74W-D1270N-CFTR, causes a more severe effect on chloride secretion and, consequently, on the phenotype [25]. The R74W and D1270N variants are already known as disease-causing [11] and are treatable with either VX-661 + VX-770 (Symdeco/Symkevi) or VX-445 + VX-661 + VX-770 (Trikafta/Kaftrio), referenced on https://www.vertextreatments.com (accessed on 20 January 2023). V201M is an uncommon, benign variant in European and African American populations. Moreover, the R74W-V201M-D1270N complex allele has been previously reported in a patient featuring a congenital bilateral absence of the vas deferens (CBAVD) and was considered to be a mild phenotype [12]; however, to our knowledge, no further characterization is available in the literature. 

As class I mutations produce a nonfunctional (if any) product, our in vivo, ex vivo and in vitro data reflect the functional response of the [R74W;V201M;D1270N] genotype. Here, we combined in vivo and ex vivo data in order to demonstrate that the complex allele compromised CFTR-dependent sweat secretion, with mean C/M sweat rate values within the range for CF patients. Moreover, the intestinal current measurements registered a compromised anion secretion, with the result for the cumulative chloride secretory response being below the value that was obtained for the non-CF group. Nevertheless, a mild response to forskolin was recorded, which suggests the presence of a residual CFTR function.

The R74W-V201M-D1270N-CFTR variant is rare and quite difficult to reproduce in experimental model systems. Predicting the in vivo outcome of their cumulative effects by evaluating the contribution of each individual mutation in model systems could be inappropriate because the presence of distinct CFTR mutations in the same allele can exert neutral, positive, or negative variations in channel function [4]. As a negative impact, the additional mutation in cis can increase exon skipping [26], be deleterious for protein maturation [25,27] and reduce the efficacy of the CFTR-directed modulators [5,28]. 

Here, we have also evaluated the in vitro response of this triple-mutant allele, R74W-V201M-D1270N, to the CFTR modulators that are currently available in the clinic (corresponding to Orkambi, Symdeco and Trikafta formulations) in human intestinal organoids following the N-of-1 trial concept. For assays using 3D organoids, taking into consideration the pharmacological recovery of the organoid area that is treated with VX-661 and VX-661 + VX-445 in the absence of forskolin became important because it allowed us to appreciate the corrective effect of these molecules on rescuing R74W-V201M-D1270N-CFTR. Indeed, the total swelling of the VX-661- and VX-661 + VX-445-treated organoids measured by FIS, with the addition of the values derived from the drug-rescued steady-state organoid area (SOA), became similar to the AUC values of the swollen rectal organoids that were treated with VX-809 + VX-770 following the response to forskolin. As the FIS rates reached a plateau, it was not possible to precisely differentiate which CFTR modulator worked best for this genotype when considering only this assay in the conditions that were tested. The integration of the results that were obtained by the electrophysiological study and the biochemical assay demonstrate the superiority of the triple treatment (Trikafta) in repairing the mutant CFTR channel. The different response of the organoids to VX-809, as compared to VX-661, in recovering the lumen area is not surprising. The corrector VX-661 seems to have a superior capability of inducing a mature form of CFTR expression compared to VX-809, which is maybe due to their different mechanisms of action [29]. Moreover, the presence of VX-445, a molecule that presents both corrector and potentiator actions [30,31,32], gives additional beneficial effects in stabilizing CFTR protein maturation and function. The CFTR recovery by the VX-661-VX-445 treatment may be sufficient to allow endogenous cAMP levels to re-establish the physiological ion exchange that is necessary to recuperate the organoid area to a level that is similar to the variants that produce milder CF phenotypes, as is evident by the clear presence of a filled lumen just after 24 h of treatment with these molecules. 

The use of patient-derived cells, in this case, has the advantage of properly representing the patient’s genomic context of expression, a condition that is not achievable using other exogenous experimental models. Additional genomic variants have been reported to influence the outcome of the therapy [33]. For example, recent studies highlight that even patients homozygous for F508del present variable clinical responses to lumacaftor–ivacaftor combination therapy, making the outcome hardly predictable [34]. These studies highlight the influence of the genome in varying the phenotype, also reflecting the drug response that is typical of CF patients. Furthermore, the presence of an important CFTR deletion (CFTRdele22_24) allows us, for the first time to our knowledge, to isolate the functional contribution of this complex allele. In this case, our results have demonstrated that VX-809, VX-661 and VX-661-VX-445 improve the function of this triple mutant, suggesting that the variant V201M does not negatively affect the responsiveness to the CFTR modulators and support the potential efficacy of the drugs that have been tested. The response to a CFTR modulator in the primary cell samples might correctly guide the clinician to a specific treatment. ICM and optical beta-adrenergic sweat tests were relevant for supporting the diagnosis and might be of use in monitoring the effect of the modulators if/when they are approved for this patient.

## 4. Conclusions

In conclusion, this case report describe for the first time the response of this complex allele to clinically available CFTR modulators and the difficulties in counseling CF patients carrying rare variants of the CFTR gene. We also underline the relevance of investigating the functional outcome that is associated with complex CFTR genotypes and the possibility, by evaluating the in vitro response to the currently available CFTR modulatory therapies, to guide the clinicians to the best possible therapy for the individual patient.

## Figures and Tables

**Figure 1 ijms-24-05199-f001:**
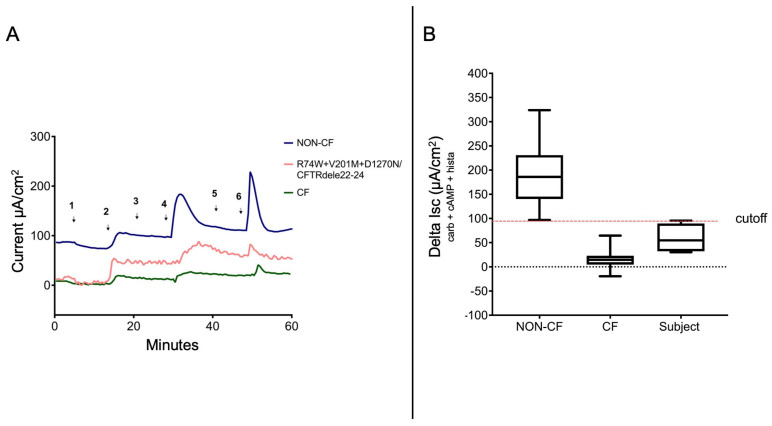
Recordings of short-circuit currents (Isc). (**A**) Representative ICM records for chloride/bicarbonate transport. Experiments were performed in the presence of 10 µM indomethacin. The arrows indicate the addition of the compounds in a standardized order to the mucosal (M) or serosal (S) side of the biopsies after 40 min of tissue equilibration: 100 µM, M, amiloride (1. Amil), 10 µM forskolin and 100 µM IBMX, M + S (2. Fsk + IBMS), 10 µM genistein M + S (3. geni), 100 µM carbachol, S (4. Cch), 200 µM DIDS, M (5. DIDS) and 500 µM histamine, S (6. hista). Blue tracings: Typical ICM tracing for chloride/bicarbonate secretion of the non-CF subject; Pink tracings: ICM tracing for chloride/bicarbonate secretion of the subject in the study; Green tracings: ICM tracing for anions secretion of the CF subject; (**B**) Value for the sum Isc (FI + cch + H) for chloride secretion obtained from the control non-CF group, CF group and the subject.

**Figure 2 ijms-24-05199-f002:**
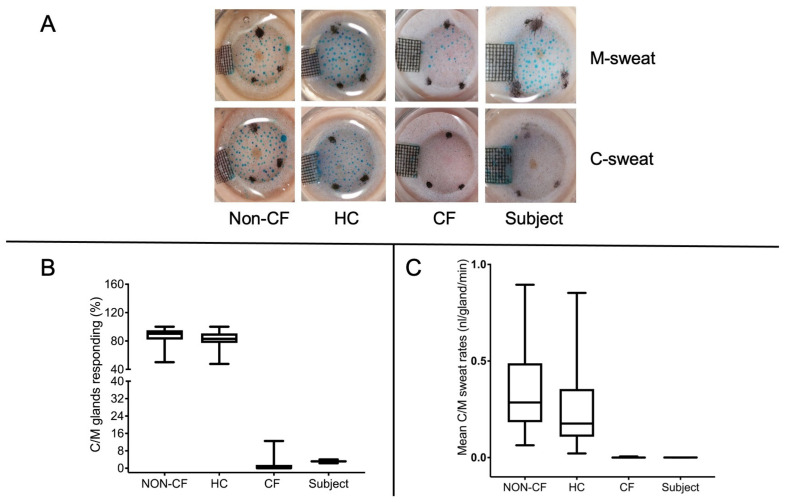
Representative image of M- and C-sweating in the p.[R74W;V201M;D1270N]/CFTR dele22_24 subject. (**A**) Each image focuses on a small region of stimulated forearm skin of the selected subjects to show blue-stained sweat droplets by single sweat glands that responded to methacholine (M-sweating) and to the β-adrenergic cocktail (C-sweating), demonstrating the activity of the CFTR channel (C-sweat); grid scale = 0.5 mm. (**B**) The percentage of glands producing C-sweat in each group (control: non-CF, healthy carriers: HC and cystic fibrosis: CF) and patient analyzed (subject). (**C**) Comparison of average C/M sweat rates measured among the same groups.

**Figure 3 ijms-24-05199-f003:**
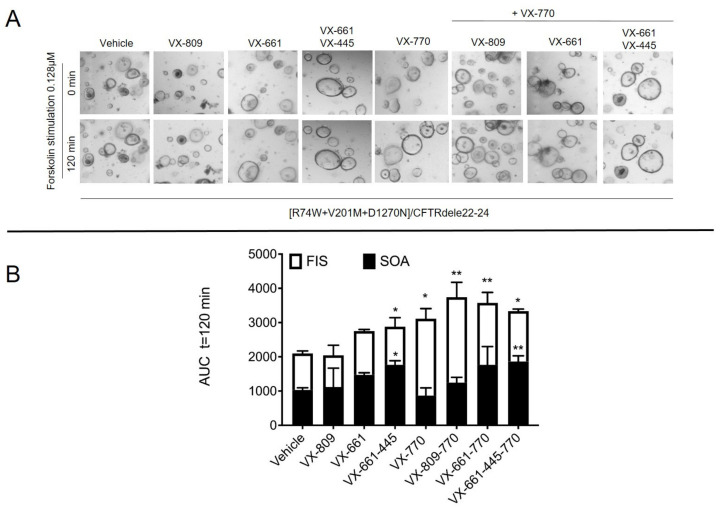
FIS rates in [R74W;V201M;D1270N]/CFTRdele22_24 3D organoids. (**A**) Bright-field microscopy images of [R74W;V201M;D1270N]/CFTRdele22_24 organoids pre-treated with correctors as a response to forskolin induction at 0.128 µM concentration alone or together with 3µM VX-770; (**B**) Swelling of rectal organoids induced by forskolin alone or in combination with VX-770. Data are expressed as the absolute area under the curve (AUC) of each duplicate for a forskolin dose of 0.128 µM and calculated from the time tracings comparable to baseline (100%, t = 120 min). FIS rates in pre-swollen organoids were adjusted by their partially recovered steady-state lumen area in the absence of forskolin (here measured as steady-state total organoid area; SOA), avoiding underestimating the effect of VX-661 on restoring CFTR function by the FIS assay. Data are means ± SD. Asterisks indicate significant difference compared with vehicle (* *p* -value< 0.05, ** *p*-value < 0.01, one-way ANOVA).

**Figure 4 ijms-24-05199-f004:**
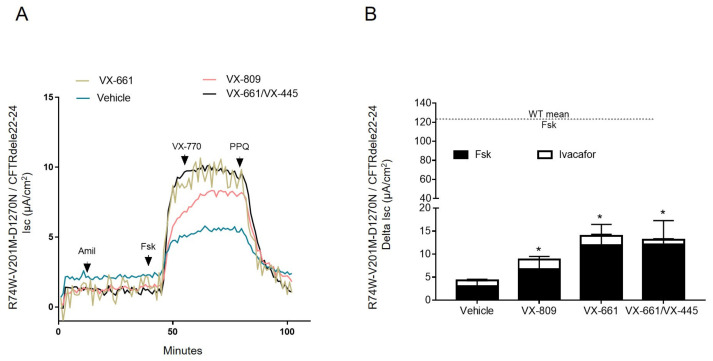
Electrophysiological response of the 2D monolayer of [R74W;V201M;D1270N]/CFTRdele22_24 colonoids. (**A**) Representative Isc tracing of anion secretion mediated by CFTR in colonoids preincubated with vehicle (dimethyl sulfoxide: DMSO) and the indicated CFTR modulators, following challenge with forskolin and ivacaftor; (**B**) Magnitude of response expressed as delta Isc response of cAMP-induced chloride currents by forskolin and ivacaftor in the 2D monolayer of rectal epithelial organoids preincubated with the indicated CFTR modulators. The dotted line indicates the mean value (123 ± 45 µA/cm^2^) of currents registered for non-CF subjects (WT) in response to forskolin, used here as a reference. Values were normalized by the surface area of the 2D filters. Data are means ± SEM from a minimum of three independent experiments. * indicates *p* < 0.05. Mann–Whitney test.

**Figure 5 ijms-24-05199-f005:**
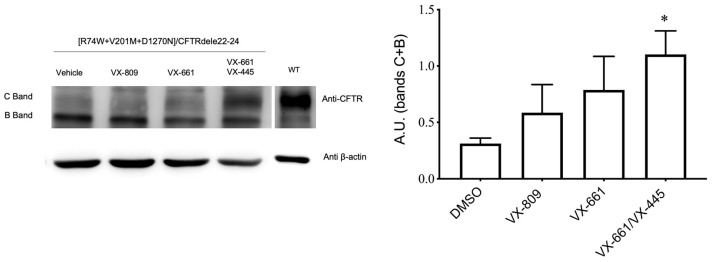
Expression of CFTR protein in [R74W;V201M;D1270N]/CFTRdele22_24 in rectal organoids. (**left**) Total protein extracts were separated on a 7.5% PAGE gel and CFTR was detected using anti-CFTR mixed mouse monoclonal antibodies (450, 570 and 596). In the CF variant, the treatment for 24 h with CFTR modulator VX-809 (3 μM) improves the expression of band C (lane 2). The combination of VX-661 (3 μM) and VX-445 (3 μM) further improves CFTR expression (lanes 3–4) as compared to untreated lysate (lane 1). A β-Actin polyclonal antibody was used as loading control; (**right**) Densitometric analysis of the results shown as representative Western blots in A. The results of the densitometric analysis shown summarized the β-actin-normalized data (C + B bands) from a minimum of three independent experiments. * *p* < 0.05 (Mann–Whitney tests).

## Data Availability

Not applicable.

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
