# Peer review of "CFTR Modulators Rescue the Activity of CFTR in Colonoids Expressing the Complex Allele p.[R74W;V201M;D1270N]/dele22_24"

_ijms, 2023, doi:10.3390/ijms24065199_

Round 1
Reviewer 1 Report
Very good
Author Response
Thank you for the note of appreciation. Please see minor adjustments in the final version.
Reviewer 2 Report
I congratulate the authors for producing a sound research. It is a great addition to better understanding the rare mutations of the CFTR
Author Response
Thank you, please see minor adjustments in the final version
Reviewer 3 Report
This paper describes the role of colonoids to investigate how CFTR modulators rescue the activities of a complex allele. The manuscript is well written, however I have a few comments for this manuscript.
1) It was very difficult to read the results and comprehend which figure relates to what part of the result because the figures corresponding to results is not mentioned in the appropriate results section.
2) Fig 3A: Could you please show what the FIS assay for VX770 looks like? It has been mentioned in the results part but is not shown in the Fig 3A.
3) Fig 4 A and B. The graphs do not make any sense with regards to the results. In each of the graphs either the healthy data or the allele data is missing.
Author Response
This paper describes the role of colonoids to investigate how CFTR modulators rescue the activities of a complex allele. The manuscript is well written, however I have a few comments for this manuscript.
1) It was very difficult to read the results and comprehend which figure relates to what part of the result because the figures corresponding to results is not mentioned in the appropriate results section.
The figures corresponding to the results are mentioned in the results section, it might be matter of editing that will be handled by editors in the final version that we expect that will be organized differently than this one .
2) Fig 3A: Could you please show what the FIS assay for VX770 looks like? It has been mentioned in the results part but is not shown in the Fig 3A.
Indeed it was not so clear, and there was a misplaced text (it moved lower than it was supposed to be) on the figure we submitted. We have slightly modify the text on the figure to indicate the presence of VX-770 in Figure 3A (see new figure 3A)
3) Fig 4 A and B. The graphs do not make any sense with regards to the results. In each of the graphs either the healthy data or the allele data is missing.
Thank you for helping us to ameliorate our presentation of the data. We have modified the legend in order to better specify the results shown. Here we show tracings from one representative experiment (panel A) and the corresponding averaged values (panel B) of the data. We also show as reference the delta Isc value (mean) derived from a group of non-CF organoids carrying wild-type CFTR (WT). we hope now the legend is easier to follow.
Reviewer 4 Report
Row 22 – the use of a bibliographic indication in the abstract is not indicated
Row 25 – translate the Italian word emoftoe in English
Row 36 – probably instead of “intro assays” could be use “in vitro assays”
Row 83 – the title “2.1. Clinical features and theratyping of p.[R74W; V201M; D1270N]/dele22_24-CFTR” is not adequate to the followed paragraph, because in this paragraph are presented only the clinical data and the results of different lab analyses.
Row 87 – translate the word Italian emoftoe in English
Row 88 – the sentence is finished by an “and” that is followed by nothing!
Rows 89 -90 – inside of parenthesis are some elements which normally should be outside it.
Rows 98-100 – probably the sentence: “p.[R74W; V201M; D1270N]/CFTR dele22_24 biopsies yielded an upward deflection after the addition of compounds (carbachol, forskolin/IBMX, and histamine) with a lower magnitude of the responses, suggesting residual CFTR function.” should be replaced with: “intestinal biopsy from the patient with p.[R74W; V201M; D1270N]/CFTR dele22_24 mutations yielded an upward deflection after the addition of compounds (carbachol, forskolin/IBMX, and histamine) with a lower magnitude of the responses, suggesting residual CFTR function.”
Row 113 – probably “if” should be replaced with “of”
Row 179 – probably the sentence: “Treatments with CFTR modulators followed led to a significant increase in…” should be replaced by: “Treatments with CFTR modulators led to a significant increase in…”
Rows 181 – 182 – Sentence: “Values for vehicle-ΔIsc is 3.1 ± 3.3, VX-770-ΔIsc: 4.3 ± 3.7; vx-809-ΔIsc: 8.8 ± 5.4, VX-661-ΔIsc: 12.8 ± 10.4 and VC-445+661-ΔIsc: 13.1 ± 10.7 µA/cm2” should be replaced with: “The results of the analyses were: vehicle-ΔIsc is 3.1 ± 3.3 µA/cm2, VX-770-ΔIsc: 4.3 ± 3.7 µA/cm2; vx-809-ΔIsc: 8.8 ± 5.4 µA/cm2, VX-661-ΔIsc: 12.8 ± 10.4 µA/cm2 and VC-445+661-ΔIsc: 13.1 ± 10.7 µA/cm2”
Row 192 – please detail the abbreviation DMSO at first use in text
Figure 5 has the legend on two pages. I suggest to modify la position of figure, so that the legend of the figure be placed completly on the same page as the figure.
Rows 519 -520 – use the international abbreviation of the journal
Row 524, 525, 544, 558-559, 563-564,569, 570, 597 – use the international abbreviation of the journal
Row 604 – use the caps lock in the name of journal
Author Response
Row 22 – the use of a bibliographic indication in the abstract is not indicated
We removed the indication of CFTR2 database
Row 25 – translate the Italian word emoftoe in English
We translated the word emoftoe in English
Row 36 – probably instead of “intro assays” could be use “in vitro assays”
Exactly, we corrected this typing error
Row 83 – the title “2.1. Clinical features and theratyping of p.[R74W; V201M; D1270N]/dele22_24-CFTR” is not adequate to the followed paragraph, because in this paragraph are presented only the clinical data and the results of different lab analyses.
We modified the original title to: Clinical features of CF subject carrying p.[R74W; V201M; D1270N]/dele22_24-CFTR mutations and functional response in vivo and ex vivo
Row 87 – translate the word Italian emoftoe in English
We did the translation of the word emoftoe in English
Row 88 – the sentence is finished by an “and” that is followed by nothing!
Sorry for the typing error, we removed the word and
Rows 89 -90 – inside of parenthesis are some elements which normally should be outside it.
We did the correction
Rows 98-100 – probably the sentence: “p.[R74W; V201M; D1270N]/CFTR dele22_24 biopsies yielded an upward deflection after the addition of compounds (carbachol, forskolin/IBMX, and histamine) with a lower magnitude of the responses, suggesting residual CFTR function.” should be replaced with: “intestinal biopsy from the patient with p.[R74W; V201M; D1270N]/CFTR dele22_24 mutations yielded an upward deflection after the addition of compounds (carbachol, forskolin/IBMX, and histamine) with a lower magnitude of the responses, suggesting residual CFTR function.”
We did modify the sentence as suggested
Row 113 – probably “if” should be replaced with “of”
We did replace the word “if” with “of”
Row 179 – probably the sentence: “Treatments with CFTR modulators followed led to a significant increase in…” should be replaced by: “Treatments with CFTR modulators led to a significant increase in…”
We correct the typing error
Rows 181 – 182 – Sentence: “Values for vehicle-ΔIsc is 3.1 ± 3.3, VX-770-ΔIsc: 4.3 ± 3.7; vx-809-ΔIsc: 8.8 ± 5.4, VX-661-ΔIsc: 12.8 ± 10.4 and VC-445+661-ΔIsc: 13.1 ± 10.7 µA/cm2” should be replaced with: “The results of the analyses were: vehicle-ΔIsc is 3.1 ± 3.3 µA/cm2, VX-770-ΔIsc: 4.3 ± 3.7 µA/cm2; vx-809-ΔIsc: 8.8 ± 5.4 µA/cm2, VX-661-ΔIsc: 12.8 ± 10.4 µA/cm2 and VC-445+661-ΔIsc: 13.1 ± 10.7 µA/cm2”
We replaced the sentence as suggested
Row 192 – please detail the abbreviation DMSO at first use in text
Done
Figure 5 has the legend on two pages. I suggest to modify la position of figure, so that the legend of the figure be placed completly on the same page as the figure.
We modified the position of the Figure 5 so that the legend fits on the same page as the figure. We expect the editors will organize the final version in order to make it more readable.
Rows 519 -520 – use the international abbreviation of the journal
Done
Row 524, 525, 544, 558-559, 563-564,569, 570, 597 – use the international abbreviation of the journal
Row 604 – use the caps lock in the name of journal
Done
Round 2
Reviewer 3 Report
Please accept in present form.